# On the Operational Utility of Measures of Multichannel EEGs

**DOI:** 10.3390/e23111434

**Published:** 2021-10-30

**Authors:** David Darmon, Tomas Watanabe, Christopher Cellucci, Paul E. Rapp

**Affiliations:** 1Department of Mathematics, Monmouth University, West Long Branch, NJ 07764, USA; david.m.darmon@gmail.com; 2Vagalume LLC, Palo Alto, CA 94306, USA; tomas@vagalume.cloud; 3Aquinas LLC, Berwyn, PA 19312, USA; cellucci@gmail.com; 4Department of Military and Emergency Medicine, Uniformed Services University, Bethesda, MD 20814, USA

**Keywords:** integrated information theory, Lempel–Ziv complexity, multichannel EEGs, electroencephalography

## Abstract

Multichannel EEGs were obtained from healthy participants in the eyes-closed no-task condition and in the eyes-open condition (where the alpha component is typically abolished). EEG dynamics in the two conditions were quantified with two related binary Lempel–Ziv measures of the first principal component, and with three measures of integrated information, including the more recently proposed integrated synergy. Both integrated information and integrated synergy with model order p=1 had greater values in the eyes-closed condition. When the model order of integrated synergy was determined with the Bayesian Information Criterion, this pattern was reversed, and in line with the other measures, integrated synergy was greater in the eyes-open condition. Eyes-open versus eyes-closed separation was quantified by calculating the between-condition effect size. The Lempel–Ziv complexity of the first principal component showed greater separation than the measures of integrated information.

## 1. Introduction

Several measures of multichannel EEGs have been explored, including measures of integrated information [1,2]. Mediano et al. [3] compared six candidate measures with computationally generated data. In this study, we investigate an objective criterion of the multichannel measures of EEGs: how effective are these measures in discriminating between different physiological states?

We assessed EEG measures by comparing values obtained in the no-task eyes-open and the no-task eyes-closed conditions. One of the most consistent properties of the EEG is alpha blocking, discovered by Berger in 1924 [4] and confirmed by Adrian and Matthews in 1934 [5]. In most individuals, but certainly not all, a very prominent alpha rhythm (8–13 Hz) is observed in the eyes-closed condition. This alpha rhythm typically disappears immediately when the eyes are opened. Examples of eyes-open versus eyes-closed spectra are presented in Hartoyo et al. [6] and Liley and Muthukumaraswamy [7].

A study comparing ten measures calculated from multichannel EEGs in the eyes-open and eyes-closed states was published by Rapp et al. [8]. The present study follows the same pattern. Three measures used in that study are included here. A more recently proposed measure of integrated information theory, integrated synergy, has been added. Additionally, this study extends the earlier study by including calculations performed with signals (both eyes-open and eyes-closed) after the alpha band had been removed with a digital filter. The efficacy of these measures to discriminate between physiological states is quantified by calculating the eyes-open versus eyes-closed effect sizes for both alpha-present and alpha-absent signals.

## 2. Participants

EEG recordings were obtained from thirteen healthy adult participants. Prior to testing, the participants gave written informed consent to take part in this research. The study reported here was approved by the Uniformed Services University Human Research Protections Program Office: Protocol DBS.2020.251. The participants were not paid or compensated for their involvement. All study procedures were conducted in accordance with human participant protections regulations as required by ethical laws and regulations set forth by the Declaration of Helsinki and the Common Rule.

## 3. Data

Free-running, no-task, monopolar EEG signals referenced to linked earlobes were obtained from awake participants in two conditions, eyes-closed and eyes-open, from $F_{Z}$, $C_{Z}$, $P_{Z}$, $O_{Z}$, $F_{3}$, $F_{4}$, $C_{3}$, $C_{4}$, $P_{3}$, and $P_{4}$ using an Electrocap. Bipolar recordings of vertical and horizontal eye movements were recorded from electrode sites above and below the right eye, and from near the outer canthi of each eye. Artifact-corrupted records were removed from the analyses. All EEG impedances were less than 5 KOhm. Signals were amplified, Gain = 18,000. Signals were digitized at 1024 Hz using a twelve-bit digitizer. Continuous artifact-free records were obtained from each subject in the two conditions. Ten-thousand-point records were used in these calculations. As reported in the introduction, an objective of this study was to determine the effect of alpha content on the resulting dynamical measures. All signals were initially passband-filtered with cut-off settings at 1 Hz and 200 Hz. A second set of signals was obtained by filtering these signals again with a Butterworth filter and an 8–13 Hz stopband.

## 4. Measures

Five measures were used in this study. The first was constructed using the Lempel–Ziv complexity [9]. Let $\left(V_{1}^{m},V_{2}^{m},\cdots ,V_{10000}^{m}\right)$ denote the mean-normalized time series of the *m*-th channel (*m* = 1, …, 10). These vectors become columns in a 10,000 ×10 matrix:(1)$$A=\left[\begin{array}{ccc} V_{1}^{1} & \cdots & V_{1}^{10}\\ \vdots & \ddots & \vdots \\ V_{10000}^{1} & \cdots & V_{10000}^{10} \end{array}\right]=V\cdot D\cdot U^{T}$$
where $V\cdot D\cdot U^{T}$ is the singular value decomposition of *A*. The singular value decomposition was calculated using the Golub–Reinsch algorithm [10,11]. *D* is the diagonal matrix of singular values $D=diag(\lambda_{1},\lambda_{2},\cdots \lambda_{10})$ where we introduce the convention $\lambda_{j}\geq \lambda_{j+1}$ for all *j*, and *U* is the corresponding orthogonal transformation. For these data, the first principal component carries more than 70% of the multichannel signal’s variance [12].

The first measure is constructed as follows. The first principal component is partitioned into a binary symbol sequence about the median, and the Lempel–Ziv complexity is calculated [9]; the pseudocode is given in Appendix A of [12].

The second measure is nearly identical to the first. In this case the mean-normalized time series of each channel is also normalized against the channel’s standard deviation before constructing matrix *A*.

The third measure is one of the earliest measures of the central nervous system information integration proposed by Tononi et al. [13]. It is constructed by comparing the degree of integration of k-dimensional subsystems with the degree of integration of the *N*-dimensional parent system. $\mathrm{Corr}(X_{j}^{k})$ is the *j*-th instance of a k×k correlation matrix formed by using *k* of the *N* channels. Tononi et al. [13] define the integration of this subsystem as follows:(2)$$I\left(X_{j}^{k}\right)=-\frac{1}{2}\ln(\det\left(\mathrm{Corr}\left(X_{j}^{k}\right)\right]$$
The average integration of the *k*-dimensional subsystem is denoted by $\langle I(X_{j}^{k})\rangle$. The system integration $C_{N}$, the third measure for this study, is determined by comparing the integration of the *N*-dimensional system $I(X^{N})$ against the integration of the subsystems of *k* channels.
(3)$$C_{N}=\sum_{k=1}^{N}\left\{\left(\frac{k-1}{N-1}\right)I\left(X^{N}\right)-\langle I\left(X_{j}^{k}\right)\rangle \right\}$$

We note that Equation (4) of Tononi et al. [13] uses k/N as the scaling factor of $I(X^{N})$. In the study performed by van Putten and Stam [14], it was argued that (k−1)/(N−1) rather than k/N is the appropriate scaling factor, which is what is used here. Pseudocode for $C_{N}$ is given in [8]. That paper also outlines difficulties with this definition of integrated information. The expression $-\frac{1}{2}\ln\left[\mathrm{Corr}\left(X_{j}^{k}\right)\right]$ has a singularity of infinite integration if two channels are completely correlated. Perfect correlation will not occur with biological data, but highly correlated signals can be observed in high-density EEG montages. Notably, in the calculations with simulation data in the study by van Putten and Stam [14], a noise term was added to the simulations to produce computationally stable examples. A numerically stable alternative definition of integration based on the Morgerra covariance complexity [15] was identified in [8].

The fourth measure examined in this study is another measure of integrated information [16,17]. Within a broad conceptual structure, a system is deemed to be “complex” if it balances integration (portions of the system work together) and segregation (portions of the system work in isolation). This is implemented by quantifying the information that the current state of the system has about its past state and comparing this information in the fully integrated system against a system partitioned to have the weakest informational links between partition elements; that is, the system partitioned to have the weakest possible integration. The latter is called the Minimum Information Partition. Broadly stated, Integrated Information Theory has been presented in three versions: Version 1 [18], Version 2 [1], and Version 3 [19]. Version 3 was formulated for discrete systems, and since our present objective is the analysis of continuous EEG signals, we focused on Version 2 implementations. Mediano, Seth, and Barrett [3] compared six measures of integrated information. In simulations on Gaussian vector autoregressive processes, they obtained best performance with three measures: integrated synergy, ψ, decoder-based integrated information, $\Phi^{*}$, and causal density, CD. Causal density is the average of the conditional transfer entropies between each pair of components of the system, and it could therefore be argued that it falls outside the domain of Integrated Information Theory. Between ψ and $\Phi^{*}$, ψ is easier to compute for in Gaussian processes. We therefore selected integrated synergy [20] for incorporation into this study. A concise mathematical description of integrated synergy is given in the Appendix A of this paper.

Measure 5 is again integrated synergy. In this case, however, the model order of the underlying Gaussian autoregressive process is not fixed at p=1 as in [3], but is determined for each multichannel data set by the Bayesian Information Criterion, BIC. Identified orders were between three and six.

## 5. Results

Five measures were obtained in two behavioral conditions (eyes-open and eyes-closed) for two-signal configurations (alpha band present and alpha band removed). An initial examination of eyes-open versus eyes-closed differences suggested that results from one participant were markedly different. A systematic investigation was undertaken, and it was established that results from that participant met a standard outlier criterion, one and one half times the interquartile range, for six of the ten difference scores (five measures obtained with the alpha present and the same five measures obtained with the alpha removed, producing a total of ten measures). This data set was removed from the analysis. the results reported here were obtained using data from the remaining twelve participants.

The values obtained from each measure in each condition are presented in the Appendix A. Of more immediate interest are the differences observed in the eyes-open versus eyes-closed conditions, as shown in Table 1. Cases where the complexity of eyes-closed is greater than eyes-open are highlighted in red. In the case of $C_{N}$, the observation that eyes-closed values are greater than eyes-open values is consistent with van Putten and Stam [14], van Cappellen van Walsum et al. [21], and Rapp et al. [8]. This ordering for $C_{N}$ was not, however, observed in Trujillo et al. [22]. This result is addressed in the discussion. The two measures using the Lempel–Ziv complexity have greater values in the eyes-open condition. When computed with a model order equal to one, integrated synergy also has a greater value in the eyes-closed condition, but this pattern is reversed when the Bayesian Information Criterion is used to identify an appropriate value of model order. For *p* chosen by BIC, the results are consistent with Lempel–Ziv, as used here, and with nine measures of the 2005 study; integrated synergy is greater in the eyes-open condition when *p* is chosen by BIC.

The difference scores were quantified by calculating the corresponding effect sizes. A standard estimator (difference normalized against the standard deviation) and a robust estimator [23] were calculated. For both estimators 10,000 bootstrap samples were used to construct 95% confidence intervals of the effect sizes using a bias-corrected and adjusted (BCa) confidence interval [24]. The results obtained with the standard estimator are in Table 1 and shown in Figure 1. Effect sizes obtained with the robust estimator are included in the Appendix A. Results where the eyes-closed values are greater than the corresponding eyes-open values are highlighted in red.

We next consider the statistical significance of the difference between the eyes-open and eyes-closed conditions. The small number of participants in the study argued against a significance test that assumed a normal distribution. The Wilcoxon signed-ranks test was used to assess the statistical significance of the differences obtained in the eyes-open versus the eyes-closed conditions with a two-sided test (p<0.05). The null hypothesis (measures obtained in the eyes-open condition are indistinguishable from measures obtained in the eyes-closed condition) was rejected in eight of the ten cases (five measures considered with and without alpha band content, thus giving ten measures). The two measures that failed to reject the null hypothesis were $C_{N}$ with the alpha content removed, and integrated synergy, ψ, model order p=1, also for the case where the alpha band was removed.

Nonparametric correlations between measures were quantified with Kendall’s tau. The correlations shown in Table 2 were calculated by combining both eyes-closed and eyes-open data in the alpha band present and alpha band absent conditions. Calculations of Kendall’s tau obtained separately with eyes-closed and eyes-open data are in the Appendix A. The 95% confidence intervals are based on the percentile bootstrap with 2000 bootstrap samples. As seen in the table, the results are largely unremarkable. In all cases, the correlation decreases with the removal of alpha. The two variants of the Lempel–Ziv complexity are highly correlated. Measure 4, which is ψ, p=1 is negatively correlated with ψ, *p* via BIC, which is expected since on average, the eyes-open versus eyes-closed relationship is reversed in the two cases.

## 6. Discussion

Four provisional conclusions follow from the computational results.

First, in the case of integrated synergy, identifying an appropriate model order is an essential element of the analysis. A comparison of the p=1 and *p* via BIC results showed that eyes-open minus eyes-closed values change sign when *p* via BIC is used, and that the *p* via BIC results align with other measures of complexity. In addition to the calculations presented here, comparisons should be made to the results in [8]. Of the ten measures examined in that study, the complexity was greater in the eyes-open condition in nine measures. The only exception was the 1994 measure of integrated information, $C_{N}$ [13]. It is concluded that a statistically responsible determination of model order for integrated synergy does not simply result in modest quantitative differences. Qualitative differences are obtained.

The signs of the $C_{N}$ eyes-open versus eyes-closed difference merit further consideration. In addition to [8], van Putten and Stam [14] examining EEGs and van Cappellen van Walsum et al. [21] examining MEGs found $C_{N}$ greater in the eyes-closed condition. Van Putten and Stam [14] also found that $C_{N}$ increased in examples of generalized seizures and severe postanoxic encephalopathy. van Cappellen van Walsum et al. [21] examined MEG records obtained from patients with a probable diagnosis of Alzheimer’s dementia according to the NINCS-ADRDA criteria [25]. They found $C_{N}$ to be higher in Alzheimer’s disease as compared to controls in the 2–4 Hz and 4–8 Hz frequency bands.

The integrated information results presented by Trujillo et al. [22] diverge from those presented here and by previous investigators. Trujillo et al. found that integrated information was greater in the eyes-open condition. Trujillo et al. analyzed signals that had been bandpassed to the theta/alpha range (4–13 Hz) and to the beta range (14–30 Hz). Additionally, the signals used in Trujillo et al. were transformed to a normal distribution using a procedure published by van Albada and Robinson [26]. When Trujillo et al. calculated integrated information with their data in the absence of this transformation, a greater value was obtained in the eyes-closed condition, as was found in our calculations of integrated information and in calculations of integrated synergy with model order p=1. Using a model order for integrated synergy determined by the Bayesian Information Criterion produced results consistent with Trujillo et al. It seems possible that the results reported by van Putten and Stam using EEGs obtained from neurological patients and the results reported by van Cappellen van Walsum et al. with MEGs obtained from participants with dementia might be revised if the data were first normal-transformed and/or analyzed with the integrated synergy and model order determined by a model selection criterion.

Second, while the correlation between the two Lempel–Ziv measures is, as expected, high, the correlation among the integrated information measures is low. The comparatively low correlations between Lempel–Ziv-based measures and $C_{N}$ and integrated synergy are, upon consideration, not surprising. They measure very different things. Lempel–Ziv complexity is not a measure of integrated information. It quantifies message compressibility. Similarly, the low correlations between $C_{N}$ and the two implementations of integrated synergy are not surprising since they are constructed on different conceptual models. In the case of $C_{N}$, integration is the difference between the sum of the entropies of individual components and the entropy of the system. The integration of a *k*-channel subsystem of the full N-channel montage is determined by a *k*-dimensional correlation matrix ([13], Page 5035). $C_{N}$ as the difference between the integration of the full N-channel system and *k*-channel subsystems. In contrast, integrated synergy quantifies information movement and uses the system’s past to predict its future. In the order-1 case, it is assumed that the most recent past is sufficient to predict the time series. In the order-*p* case, where *p* is greater than one, an expanded history is used. A mathematical presentation of integrated synergy is given in the Appendix A.

Third, the examination of differences in the eyes-open and eyes-closed conditions failed to meet statistical significance for Measure 3, $C_{N}$, and for Measure 5, integrated synergy with *p* via BIC, when the alpha band content was removed from the EEG. The failure of $C_{N}$ to show a statistically significant between-condition effect size when the alpha content was removed can perhaps be understood by recalling that, as outlined above, it is based on measures of between-channel correlation that would be reduced when the alpha component is removed. The result observed in the absence of alpha content with integrated synergy for the case where order *p* is determined by the Bayesian Information Criterion presents a greater puzzle. If p=1, the estimated effect size is statistically significant. We cannot explain why significance was lost when a statistically responsible procedure was used to determine model order.

Fourth, in a comparison of effects sizes, the Lempel–Ziv complexity of the first principal component of a multichannel EEG was more effective in discriminating between the two conditions than $C_{N}$ or integrated synergy. We have no immediate reason to know why this is the case, and it would be inappropriate to speculate based on a study with twelve subjects and a low-density montage. Lempel–Ziv complexity is, however, known to be robust to noise. This may be a contributory factor. A study comparing the sensitivity of these measures to noise might be informative.

## Figures and Tables

**Figure 1 entropy-23-01434-f001:**
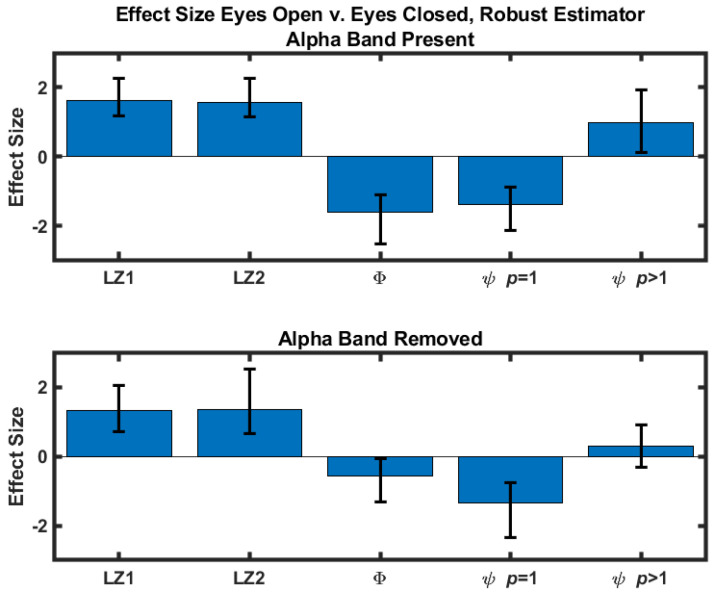
Eyes-open versus eyes-closed effect size as quantified by the standard estimator $\left({\hat{\mu }}_{\mathrm{open}}-{\hat{\mu }}_{\mathrm{closed}}\right)/{\hat{\sigma }}_{\mathrm{open}-\mathrm{closed}}$ for five measures. Note that the effect size estimates are unitless. Confidence intervals were determined with a bias-corrected and adjusted bootstrap. (**Top**) Effect sizes calculated with signals that contain the alpha component. (**Bottom**) Effect sizes calculated after the alpha component had been removed with an 8–13 Hz stopband filter.

**Table 1 entropy-23-01434-t001:** The sample averages and standard deviations of the difference scores for measures between eyes-open and eyes-closed, estimates of $\left(\mu_{\mathrm{open}}-\mu_{\mathrm{closed}}\right)/\sigma_{\mathrm{open}-\mathrm{closed}}$, and 95% confidence intervals for the effect size without (top CI) and with (bottom CI) Bonferroni correction for 5 comparisons. (Top) Measures calculated with signals that contain the alpha component. (Bottom) Measures calculated after the alpha component had been removed with an 8–13 Hz stopband filter.

Measures	Eyes-Open—Eyes-Closed	Estimated	95% CI
(Alpha Band Present)	Mean ± StDev	Effect Size	
Binary Lempel–Ziv	63.4167	1.63	(1.17, 2.26)
Signals mean normalized	± 36.9360		(1.07, 2.53)
Binary Lempel–Ziv	51.2500	1.56	(1.13, 2.27)
Signals normalized by mean	±31.5873		(1.03, 2.61)
and by standard deviation			
$C_{N}$	−0.6469	−1.60	(−2.53, −1.12)
Tononi et al., 1994	±0.3973		(−3.01, −0.99)
Equation (4)			
ψ p=1	−0.8556	−1.40	(−2.13, −0.90)
Mediano et al., 2019	±0.5906		(−2.42, −0.76)
Equation (23)			
ψ*p* via BIC	0.2013	0.99	(0.13, 1.93)
Mediano et al., 2019	±0.1944		(−0.25, 2.15)
Equation (23)			
**Measures**	**Eyes-Open—Eyes-Closed**	**Estimated**	**95% CI**
**(Alpha Band Removed)**	**Mean ± StDev**	**Effect Size**	
Binary Lempel–Ziv	42.7500	1.33	(0.72, 2.05)
Signals mean normalized	±30.8460		(0.53, 2.3)
Binary Lempel–Ziv	39.6667	1.35	(0.65, 2.52)
Signals normalized by mean	±28.2081		(0.43, 3.25)
and by standard deviation			
$C_{N}$	−0.2102	−0.57	(−1.33, −0.52)
Tononi et al., 1994	±0.3658		(−1.34, 0.22)
Equation (4)			
ψ p=1	−0.6755	−1.34	(−2.35, −0.75)
Mediano et al., 2019	±0.4900		(−2.90, −0.60)
Equation (23)			
ψ*p* via BIC	0.0679	0.31	(−0.32, 0.90)
Mediano et al., 2019	±0.2093		(−0.65, 1.14)
Equation (23)			

**Table 2 entropy-23-01434-t002:** Estimates and 95% bootstrap confidence intervals for Kendall’s tau within-subject, between measures in the eyes-open and eyes-closed conditions. (Top) Measures calculated with signals that contain the alpha component. (Bottom) Measures calculated after the alpha component had been removed with an 8–13 Hz stopband filter.

Alpha Band	LZ1	LZ2	$C_{N}$	ψ p=1	ψ*p* via BIC
Present					
LZ1	–	0.8436	−0.4218	−0.3636	0.4000
		(0.718, 0.945)	(−0.612, −0.202)	(−0.613, −0.076)	(0.102, 0.654)
LZ2		–	−0.4058	−0.4130	0.3406
			(−0.638, −0.152)	(−0.645, −0.116)	(0.065, 0.58)
$C_{N}$			–	0.2246	−0.3351
				(−0.09, 0.523)	(−0.670, −0.015)
ψ p=1				–	−0.4203
					(−0.668, −0.129)
**Alpha Band**	**LZ1**	**LZ2**	$C_{N}$	ψ p=1	ψ p **via BIC**
**Removed**					
LZ1	–	0.6374	−0.0873	−0.1673	0.1164
		(0.349, 0.831)	(−0.387, 0.25)	(−0.468, 0.157)	(−0.130, 0.356)
LZ2		–	−0.2190	−0.1971	0.1679
			(−0.543, 0.142)	(−0.455, 0.085)	(−0.099, 0.425)
$C_{N}$			–	0.1522	−0.0435
				(−0.5, 0.184)	(−0.346, 0.269)
ψ p=1				–	−0.2971
					(−0.549, −0.019)

## Data Availability

Requests for data should be directed to the corresponding author. Availability is subject to Department of Defense policies concerning human research data.

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
