# Peer review of "On the Operational Utility of Measures of Multichannel EEGs"

_entropy, 2021, doi:10.3390/e23111434_

Round 1
Reviewer 1 Report
In this manuscript, Darmon et al. investigate three supposedly measures of integrated information (Lempel-ziv complexity of the first principal component, Tononi's neural complexity, and Griffith's Integrated Synergy) on EEG signals obtained from 13 healthy subjects in two behavioral conditions: eyes open and eyes closed. The results show variable performances of the measures in discriminating between these states. The authors then argue that their findings encourage caution when advocating for the use of measures of integrated information as measures of consciousness.
While the manuscript is generally well-written, the methods are adequately described and the results are clear enough, I believe this study is of very limited interest, for the reasons below.
As it is clear from the title, the main goal of this study is to evaluate "the operational utility" of measures of integrated information. In this regard, although neural complexity (Cn) was indeed introduced by Tononi et al. to quantify the coexistence of functional differentiation (information) and functional integration in the brain, Cn is based on mutual information and thus reflects statistical dependencies between thalamocortical circuits rather than causal interactions. This is important because causality is a crucial element of the Integrated Information Theory of consciousness (IIT): IIT equates the "quantity of consciousness" with the "irreducible cause–effect power" of the system (Tononi 2016, doi:10.1038/nrn.2016.44). Moreover, this is arguably one of the reasons why Tononi has abandoned Cn for PHI, which is a measure based on effective rather than mutual information (see for instance Tononi and Sporns 2003, doi:10.1186/1471-2202-4-31 as well as Tononi 2001 doi:10.4449/aib.v139i4.510). I thus find the results of Cn reported in the present study (including the considerations about the signs of the Cn eyes-open versus eyes-closed conditions) quite irrelevant for this particular topic: we already know that Cn is a poor approximation of the type of integrated information that is relevant for consciousness. At the same time, the Lempel-Ziv complexity of the principal component of spontaneous ongoing EEG oscillations can hardly be considered as a measure of Integrated Information at all. Therefore, from the three "measures of integrated information" employed in this study, only the third one, Integrated Synergy, seems relevant here and I think it is safe to say that this study is essentially about the operational utility of Integrated Synergy as a measure of integrated information.
The same conclusion can be drawn from considerations about the novelty of this work. As stated in the Introduction, the authors have already reported, 16 years ago, on the effectiveness of Lempel-ziv complexity and Tononi's neural complexity in discriminating between eyes open and eyes closed resting states. Compared to their previous work, the present study innovates by incorporating a third measure - Integrated Synergy - and by including the calculation of these measures after removing the alpha band from EEG signals. But the latter procedure has just confirmed what could be trivially expected: removing the alpha band produces a reduction of effect sizes in discriminating between eyes open and eyes closed. Thus, again, the main substantial innovation of the present study is the use of Integrated Synergy.
In this context, the reported values of Integrated Synergy for different criteria of model order are certainly the most interesting result of this work. But I don't think these findings are sufficiently relevant to justify publication. In particular, the data reported here are certainly not enough to "evaluate the operational utility" of Integrated Synergy as a measure of integrated information. IIT is a theory of consciousness and evaluating the operational (practical) utility of measures inspired on IIT, such as Integrated Synergy, depends on evaluating these measures across different states of consciousness (mainly global states, although local states could also be investigated if done carefully - see Bayne et al 2016 for more on this distiction https://doi.org/10.1016/j.tics.2016.03.009 ). Given that there is no information about the global state of consciousness of the subjects who participated in this study (see also below), I assume here that recordings were all obtained during wakefulness, i. e., all subjects in both conditions were awake and conscious, which is probably the reason why the authors write that, instead of trying to quantify consciousness, they are addressing in this study a "more pedestrian but answerable question: is this measure useful?" (lines 222-223). But I'm afraid this more "pedestrian" question is also an ill-defined one: useful for what? The authors seem to believe that discriminating eyes closed from eyes open is a necessary criterion for the quantification of consciousness - or, as they write, it is "a first primitive criterion" for the utility of such measures. But is it? Should a measure of the brain's capacity for consciousness depend on whether alert subjects are lying with their eyes opened or closed? I don't think so. First, it is clear that these behavioral conditions (eyes open/closed) can be dissociated from the capacity for consciousness - in particular, patients suffering from the now so called "unresponsive wakefulness syndrome" may be unconscious while awake and with eyes open. Second, empirical measures of consciousness may well distinguish between conscious states while being insensitive to such behavioral conditions. Consider, for instance, the Perturbational Complexity Index (PCI), an empirical measure of consciousness inspired by IIT (Casali et al., 2013 doi: 10.1126/scitranslmed.3006294) that has gained considerable attention in recent years. PCI attained such a high accuracy in discriminating conscious from unconscious individuals (Casarotto et al., 2016 doi:10.1002/ana.24779) that it was recently recommended in a number of expert reviews and clinical guidelines for the diagnosis of disorder of consciousness (Giacino et al., 2018 doi: 10.1212/WNL.0000000000005926; Kondziella et al., 2020 doi: 10.1111/ene.14151, Bai et al., 2020 doi: 10.1007/s00415-020-10095-z; Comanducci et al., 2020 doi: 10.1016/j.clinph.2020.07.015). At the same time, PCI does not discriminate eyes open from eyes closed in awake subjects (Casali et al., 2013 doi: 10.1126/scitranslmed.3006294). And why should it if the goal here is to objectively asses the capacity for consciousness independent of behavior? Indeed, one may even argue in the opposite direction of Darmon et al.: measures of integrated information that are relevant for consciousness should not depend on behavioral states that can be dissociated from consciousness (such as eyes open/eyes closed conditions).
In any case, it seems to me that one can only employ the results of the present work in evaluating the operational utility of measures of consciousness if the variability observed across different behavioral conditions for a fixed global state of consciousness (as reported here) is compared to the variability observed across different states of consciousness for a fixed behavioral condition. Unfortunately, this latter step, crucially important for the main conclusion, is missing in this study. An for this reason I do not believe that the main conclusion of this work is (or can be) supported by the data presented here.
Other issues:
- Lines 51-52 "Free-running, no-task, monopolar EEG signals referenced to linked earlobes were obtained in two conditions, eyes closed and eyes open". Did the authors control for fluctuations in the level of arousal (wakefulness/sleep) during the experiment? (Were subjects awake during the experiment?)
- I wonder what argument the authors have used for including the Binary Lempel-Ziv complexity of the principal component of spontaneous (ongoing) EEG oscillations as a "measure of EEG integrated information" while excluding Causal Density (line 109). The former measure seems farther from the domain of IIT than the latter, as Causal Density at least tries to estimate causality, a key factor for IIT. Could the authors please explain this?
- I think the presentation of results could be improved by reducing the number of tables to two (alpha and no alpha, which means merging tables 1 and 3, and tables 2 and 4) as well as by adding the p-values of wilcoxon's ranksum tests to the tables.
- It seems to me that the text has been written in a hurry. Some paragraphs could be much improved in terms of style (as the paragraph on integrated synergy, line 91, which starts rather vaguely), or content (for instance, the paragraphs starting on lines 212 and 214 are just restatements of the results and add no value to the discussion). There are also some minor spell errors (for instance, check lines 95, 146, 158-159).
Reviewer 2 Report
In this manuscript, the authors present a study aimed to operationally compare measures of integrated information applied to human electroenphalographic data (EEG). The authors use data from 12 subjects recorded using 10 electrodes during two resting-state conditions: with eyes open and with eyes closed. The data is clean using standard procedures. Five different measures of complexity and information integration are then computed on the EEG data with and without alpha frequency components. Finally, the authors report on the utility of each measure to differentiate between conditions and the relation among the metrics.
As a contributor to the field of consciousness, using EEG-based measures to differentiate between diverse pathological and physiological states of consciousness, I find this manuscript touches a relevant issue of the field. When dealing with measures of complexity/information integration, which one to use?
However, I do believe that this manuscript could be improved, providing the reader with relevant information regarding this pertinent question. In its current form, it is hard to see what the authors intend to demonstrate.
The following are a series of comments and questions on each individual section of the manuscript.
Introduction
- It is true that there is a theoretical and empirical link between measures of integrated information and consciousness. But why the authors decide to apply measures for consciousness to open/closed eyes resting state? Perhaps there is no need to mention consciousness as the goal and analysis of the manuscript does not deal with this complex topic, rather than two different physiological states.
- In the data section, the objective is later stated as “determine the effect of alpha content on the resulting dynamical measures”. I believe that this is not clear in the introduction. Furthermore, the title should also reflect this fact, as this is indeed the actual utility being evaluated. If the authors wish to evaluate the utility of this measures for different cognitive/physiological states, the authors could rely on open databases such as the HCP dataset with contains resting state as well as different active paradigms.
Measures:
- Introduction states that this study follows up a previous publication [1] in which 10 measures are compared, incorporating one more, but only 5 measures are described. Please rephrase or clarify. Also, using the same names as in the previous article will pinpoint clearly which is the added measure and its associated results. Furthermore, the names should match the measure. As an example, measure #1 is described as the Binary Lempel-Ziv complexity of the first principal component, but in table 1 it is mentioned as “Binary Lempel Ziv, Signal mean normalized”.
Results:
- The text describes the results, but just referring to tables, supplementary material and the discussion. Readability could be improved by summarizing the obtained results.
- Table 1 and Table 2 could be summarized into one figure, showing the individual values for each subject as a swarm plot with a boxplot overlayed or a raincloud plot. Examples here: https://seaborn.pydata.org/generated/seaborn.swarmplot.html
- Were the statistical tests corrected for multiple comparisons? Can the authors show the p-values of each test with and without correction for multiple comparisons?
- Given the small sample size, using bootstrap to estimate the effect size might yield over estimated values. The confidence intervals for the effect sizes are quite broad. Since the authors use Wilcoxon Signed rank tests, why not report the effect size from the statistical test? See [2], [3]
- Data from tables 5 and 6 can also be displayed as figures showing the individual correlations for each pair of metrics. Reporting the tau-coefficient is not enough. Can the authors indicate the associated p-values of each tests?
Discussion
- The authors conclude that the parameter p of the integrated synergy measure can drastically change the results. However, this is hard to conclude from the results as the authors present them. Just showing that the sign of the correlation is reversed is not enough, given that samples are paired and each measure is computed for every sample. The results should include analysis showing exactly this information.
- Given that this study follows the same analysis and measures as in [1], what is the relation between the results obtained here and the previous results? Given that this study is strongly linked to the previous study, a table comparing the obtained will improve readability. In this current form, the reader needs to refer to the previous work to understand the references, yielding a non-self-contained manuscript.
- The same as in the previous comments apply to other studies using the same metrics. The authors do a review of previously obtained results in the discussion. A table summarizing the finds can improve readability and interpretation.
- The authors conclude that correlations between two LZ measures is expected to be high, but between II measures are low. Can you discuss why is this happening? Indeed, out of the 4 principal conclusions, only the first one is discussed.
- Discussing the usefulness of a metric defined to quantify consciousness in a study which analysis two conditions in which consciousness does not vary is out of context. The authors discuss if a measure constructed to quantify consciousness is useful by testing its capacity to distinguish different cognitive states. The current manuscript only analyses data from resting state with eyes open and closed, which are not necessarily two different cognitive states, but two different physiological states. Please restrain the analysis and discussion to the eyes open/closed task.
I believe that with these modifications, this manuscript could provide more evidence supporting the utility of integrated synergy as a measure of information integration, as well as the importance of correctly parametrising it.
References:
[1] P. E. Rapp, C. J. Cellucci, T. A. A. Watanabe, and A. M. Albano, “QUANTITATIVE CHARACTERIZATION OF THE COMPLEXITY OF MULTICHANNEL HUMAN EEGS,” Int. J. Bifurc. Chaos, vol. 15, no. 05, pp. 1737–1744, May 2005, doi: 10.1142/S0218127405012764.
[2] R. Rosenthal and D. B. Rubin, “requivalent: A Simple Effect Size Indicator,” p. 7.
[3] C. O. Fritz, P. E. Morris, and J. J. Richler, “Effect Size Estimates: Current Use, Calculations, and Interpretation,” p. 19.
Round 2
Reviewer 1 Report
All of my concerns were addressed by the authors when they changed the focus of the manuscript from "measures of integrated information" and "consciousness" to a study of between-state effect sizes, independently of consciousness.
I have just a couple of comments on the authors' rebuttal letter. But there is no need to respond to these comments. I now support the publication of this article.
---
Authors: "It is not suggested that the Lempel-Ziv complexity of the first principal component is a measure of integrated information theory."
Well, I'm afraid one could read the previous title of the manuscript and think this was indeed being suggested.
Authors: "The effort to find quantitative measures of change similarly motivated to construction of the Perturbational Complexity Index, PCI, cited by the Reviewer. The Review cites six papers that discuss PCI. We are familiar with the work done by Massimini and his colleagues and cited Casali, et al. in an earlier publication. As the Reviewer notes, Casali, et al found that PCI did not discriminate between eyes-open and eyes-closed in awake subjects. Would it be reasonable to suppose that PCI, the clinical utility of which is not in dispute, would be sensitive to variations of cognitive engagement associated with eyes-closed contra eyes-open in awake healthy participants?"
I was not interested in whether the authors have cited or not the work of Massimini in their publications. I don't think this is relevant here at all. The reason I cited these papers in my review was to illustrate the crucial issue of the previous version of the manuscript, which was also pointed out by the second reviewer: the confusion between "capacity for consciousness" (or "level of consciousness", or "global states of consciousness", if you like) and "content of consciousness" (or "local states of consciousness"). PCI is an example of a measure that appears to be quite sensitive to the former while being insensitive to the later. Anyway, as consciousness is not the focus of this revised version, this is not an issue anymore.
Authors: "These papers are cited in the revision".
Again, I don't understand the relevance of this comment. The reason I cited the papers of Tononi et al. was to support my argument. I was not asking the authors to cite them in their manuscript.
Authors: "As previously noted, an objective of the exercise was to compare the between-state discrimination obtained with some integrated information theory measures against other measures of multichannel signals. Normalized Lempel-Ziv complexity is widely used, and this encouraged its incorporation into the study. It can be noted, for example, that the measure utilized in the calculation of the Perturbational Complexity Index is a calculation of the normalized Lempel-Ziv complexity of the binary SS(x,t) matrix. Causal density was considered as were several other candidate measures. For example, Tegmark (PLoS Computational Biology) identified hundreds of IIT options. In a study of this kind, choices must be made."
Certainly, I agree that choices must be made. But in the previous version of this study, which was focused on "measures of integrated information", these choices appeared to be quite arbitrary in my opinion. In this respect, one final observation regarding integrated information measures and other measures of multichannel signals: calculating Lempel-Ziv complexity of the binary SS matrix (thus after perturbing the brain, averaging across epochs, and statistically comparing pre and post stimulus activities) has a completely different meaning than calculating Lempel-Ziv complexity of ongoing (spontaneous) EEG oscillations. Causality is the key element here, not the complexity metric.
Reviewer 2 Report
All my issues were addressed when the authors switched from integrated information and consciousness to a simple but thorough analysis of these measures with regards to eyes open vs eyes closed. In its current version, the manuscript is well framed in the scope of the analysis and self-contained.
My only comment is regarding the introduction. The first paragraph poses the question on discriminating cognitive/physiological states, but cognitive states are not assessed. In paragraph three, the authors just refer, correctly, as physiological states.
There is no need for another round of reviews. Just this modification.
Reviewed by Dr. Federico Raimondo.